# Feasibility Study of Electrified Light-Sport Aircraft Powertrains †

**Madeline McQueen [1], Ahmet E. Karataş [1,\*], Götz Bramesfeld [1], Eda Demir [1] and Osvaldo Arenas [2]**

[1] Department of Aerospace Engineering, Ryerson University, 350 Victoria St., Toronto, ON M5B 2K3, Canada; madeline.mcqueen@ryerson.ca (M.M.); bramesfeld@ryerson.ca (G.B.); eda.demir@ryerson.ca (E.D.)

[2] Gas Turbine Laboratory, Aerospace Research Centre, National Research Council, 1200 Montreal Rd., Ottawa, ON K1A 0R6, Canada; osvaldo.arenas@nrc-cnrc.gc.ca

\* Correspondence: karatas@ryerson.ca

† This paper is an extended version of our paper published in Proceedings of the AIAA SciTech 2022 Forum, San Diego, CA, USA, 3–7 January 2022.

**Abstract:** A theory-based aerodynamic model developed and applied to electrified powertrain configurations was intended to analyze the feasibility of implementing fully electric and serial hybrid electric propulsion in light-sport aircraft. The range was selected as the primary indicator of feasibility. A MATLAB/Simulink environment was utilized to create the models, involving the combination of proportional-integral-derivative controllers, aerodynamic properties of a reference aircraft, and powertrain limitations taken from off-the-shelf components. Simulations conducted by varying missions, batteries, fuel mass, and energy distribution methods provided results showcasing the feasibility of electrified propulsion with current technology. Results showed that the fully electric aircraft range was only 5% of a traditionally powered aircraft with current battery technology. Hybrid electric aircraft could achieve 44% of the range of a traditionally powered aircraft, but this result was found to be almost wholly related to fuel mass. Hybrid electric powertrains utilizing an energy distribution with their optimal degree of hybridization can achieve ranges up to 3% more than the same powertrain utilizing a different energy distribution. Results suggest that improvements in the power-to-weight ratio of the existing battery technology are required before electrified propulsion becomes a contender in the light-sport aircraft segment.

**Keywords:** electrification; hybrid; powertrain; simulation; electric propulsion; degree of hybridization; flight performance



## 1. Introduction

Climate change is a significant threat to Earth and is mainly caused by humans' activities and dependence on fossil fuels. The International Energy Agency predicts that the worldwide energy demand may return to pre-pandemic levels as early as 2023 and could increase by up to 9% by 2030 [1]. Therefore, the pollutants causing climate change continue to be a significant concern. The continually increasing emissions have inspired many governments worldwide to pass rigorous regulations in all industries, including aviation. In October 2021, the chief technology officers of seven of the world's foremost aviation manufacturers pledged their commitment to making the aviation industry more sustainable by delivering technical solutions and maturing novel technologies to enable a net-zero carbon industry [2]. In addition, the US Environmental Protection Agency recently authorized primary greenhouse gas (GHG) emissions standards for aviation [3]. Furthermore, the *Canadian Airworthiness Manual* was modified to include $CO_2$ emissions practices [4].

Aviation is currently responsible for 2.4% of carbon dioxide emissions worldwide [5], and continually sees an increase in both passenger and freight transport demand. The total air passenger traffic, the number of passengers recorded on scheduled flights, has increased by a factor of three throughout the last 20 years. Additionally, passenger kilometers, the

number of passengers multiplied by the distance travelled, have risen by 6.7% throughout the last five years [6]. Lastly, the total air cargo freight weight has increased by approximately 50% worldwide over the past 15 years [7]. These worrisome values have secured commercial aviation's position as the fastest-growing source of GHG, soot, and nitrogen oxide (NOX) emissions among all industries, despite significant engine efficiency, bypass ratio, and compression ratio improvements. These improvements can be seen because carbon emissions per passenger kilometer have decreased by approximately 50% in the last twenty years [6]. The technology improvements in the industry led to a carbon footprint of about 2.5% less than the industry's growth rate. However, these trends still represent an unsustainable future.

The aviation industry's emissions concerns are different from other industries since most emissions are released at higher altitudes, resulting in a combination of direct and indirect mechanisms impacting the climate. The most apparent direct mechanism is the greenhouse effect. Once carbon dioxide is emitted, it can remain in the atmosphere for centuries. In addition, another significant aviation emission class, $NO_X$ emissions, increase tropospheric ozone concentration when emitted at high altitudes, thus warming the atmosphere [8]. Aerosol soot can eventually settle on arctic regions. This settling blackens the low albedo surface, alone causing an estimate of 25% of global warming [9]. The total global warming contribution of the aviation industry is predicted to be about 5% [8], highlighting the need to hasten improvement in low/no emission technologies.

The ideal method to help eradicate aircraft emissions is the introduction of fully electric aircraft to the industry. Unfortunately, the current level of battery energy density may not be sufficient. Batteries currently have nearly 50 times lower specific energy than traditional fuels [10]. Thus, this makes the packs required for aircraft large and heavy, with a lower than desired energy storage capability. Alternately, hybrid electric systems use both batteries and a fuel-based energy source, which is more beneficial for achieving longer overall ranges than fully electric. Thus, the development of hybrid electric aircraft can benefit the electrification goal of pollutant reduction and its limitation of aircraft range capability.

### 1.1. Overview of Hybrid Electric Propulsion

Hybrid electric propulsion uses both fossil fuels and electric power to operate an aircraft, where the goal is to reduce fuel burn, emissions, and noise. Hybrid electric aircraft can have two propulsion configurations: serial hybrid and parallel hybrid. Serial hybrid systems involve the energy sources merging through an electrical connection to power the electric motor. In this case, the only provider of power to the propeller is the electric motor. Two sources of mechanical power are combined through a mechanical connection in a parallel hybrid system where a transmission system allows both sources to provide power to the propeller. A graphical depiction of both configurations is given in Figure 1. This paper only discusses the serial hybrid configuration.

Another crucial concept to implementing hybrid propulsion is the degree of hybridization (DOH). Hybrid electric propulsion researchers have argued that a single descriptor cannot represent a complete description of the system's DOH, and it should instead be represented with two parameters [11]. These two chosen parameters represent the useful power and useful energy in a hybrid electric propulsion system. They are defined as the ratio of power or energy produced by the electric motor to the total power or energy of the propulsion system. A high DOH is environmentally beneficial but is often not feasible with present-day battery technology.

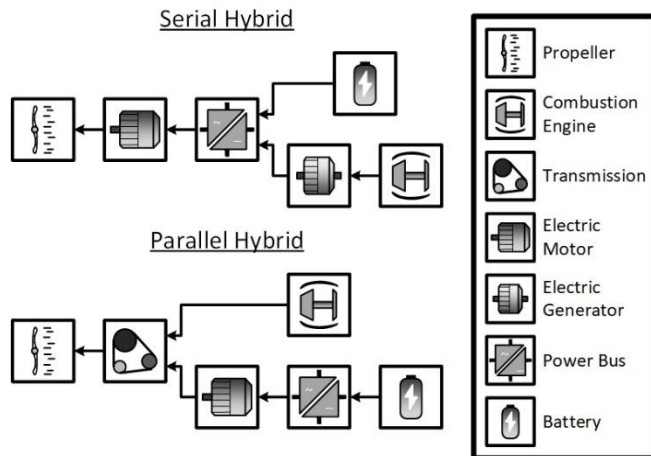

**Figure 1.** Depiction of serial and parallel hybrid electric system components and functionalities.

Several studies have been conducted on hybrid electric propulsion since it is an emerging technology, but they have been mostly theoretical. Most of these studies, e.g., [12–14], focus on applying hybrid propulsion to unmanned aerial vehicles (UAVs). The technology seems that it will be most applicable for UAVs in the next few years because of their reduced size and weight. However, a recent review paper [15] suggests that general aviation and light-sport aircraft electrification research is most critical for further electrification of the aviation industry. Likewise, Finger et al. [16] examined how the reduction of takeoff mass and energy consumption for a parallel hybrid system is affected differently for four aircraft types. Findings suggest that hybrid electric propulsion systems are viable for aircraft design requirements with short-span power and range [16].

Ludowicy et al. [17] used a mathematical modelling approach to investigate whether "light" serial hybrid aircraft will reduce fuel burn and weight compared to a traditionally powered one. Surprisingly, though serial hybrid aircraft tend to have approximately 20% more mass than traditional designs, they still allow for notable fuel savings and are deemed a usable option for future aircraft [17]. Another article investigating fuel savings using hybrid UAVs determined that 6% of fuel savings are achievable [18].

It is thought that hybrid propulsion would be impractical when utilized in larger aircraft with current technology. Pornet and Isikveren [11] attempted to prove or disprove this claim in a narrow body transport aircraft. Notable fuel savings were achieved when high DOHs were tested, but the technology is limited by battery energy density and, hence, is limited to low-range designs. The success of hybrid electric propulsion is considerably dependent on utilizing the synergies among distributed electric propulsion, aerodynamics, and structures [11,19–21].

Apart from the theoretical studies, initial modelling and simulation analyses regarding electrified propulsion have also been completed in the literature. Some examples include: solely analyzing the propulsion system, analyzing the propulsion system with the addition of aerodynamic effects of the aircraft body, and determining the optimal power distribution technique between the battery and the fuel source.

Researchers based out of Georgia Institute of Technology have developed an aircraft model called "GT-HEAT", which uses a "Numerical Propulsion System Simulation", or NPSS [22]. The NPSS has heightened accuracy in propulsion and electronics but does not include abilities to examine aerodynamics and structure [10,22]. Friedrich and Robertson [12] modelled a parallel hybrid propulsion system using three main modules: navigation, propulsion system, and weight calculation. The modules used proportional-integral-derivative (PID) control and an aerodynamic model from the *X-Plane* database. A rule-based controller accomplishes the model's power distribution, with several conditions relating to the battery's instantaneous state of charge (SOC) and the aircraft's instantaneous power demand [12]. Results showed that the light aircraft model achieved fuel savings of

37% and energy savings of 30% [12]. Furthermore, by scaling the model up to analyze a 50-ton airliner, 10% fuel savings and 1.3% energy savings were found [12]. These results prove that utilizing current technologies makes hybrid propulsion more impractical in larger aircraft. Rather than a rule-based controller method, both Hung and Gonzalez [18] and Xie et al. [23] approached their energy distribution method by operating the internal combustion engine (ICE) about its ideal operating line (IOL). The IOL represents the torque-speed combinations for minimum fuel consumption. Therefore, using its guidance will maximize range while minimizing fuel consumption.

The collaboration of the German Aerospace Center (DLR) and two European universities led to a project known as HYPSTAIR. This project involves developing and analyzing hybrid propulsion components using simulations and physical testing. The physical testing rig included a small ICE: the Rotax 914 and two 13 kWh batteries [24]. The simulation environment built for HYPSTAIR, known as *HyPSim* [25], is based in MATLAB/Simulink but receives continually updated aerodynamic data from *X-Plane*. The models calculate the energy consumption and predict flight endurance using various control modules [25]. The physical testing and simulations showed that a hybrid powertrain's weight penalty decreases as the required range increases due to the fuel savings [24]. It was also found that utilizing a hybrid system improved takeoff performance compared to a conventionally powered competitor [24].

An overview of the electrified propulsion systems currently in development is given in [10]. The power outputs range between 13.5 kW and 260 kW, and the maximum takeoff masses (MTOM) range from 235 kg to 1500 kg. Therefore, the aircraft models tested in this study will also remain within these limits.

### 1.2. Research Objectives

This study involves the development of a full-scale theory-based aircraft flight-performance model. The model determines the power requirements of a light-sport aircraft (LSA) powered with either a fully electric or hybrid electric powertrain to study the feasibility of electrified aircraft in the short term. The aircraft range was chosen as the primary indicator of feasibility because of LSA users' range expectations. If the electrified aircraft's range does not meet these expectations, many users may be skeptical of utilizing electrification technology despite the various benefits, including low/no emissions. The aircraft used in this study was loosely based on a Pipistrel Virus 912. The range was calculated for several configurations by modifying three powertrain parameters: altitude, battery properties, and DOH (for the hybrid powertrain).

Cruising altitude's effect on the range is dependent on air density, thus affecting the power output. The maximum altitude is thus dependent on the maximum power output of a propulsion system, which is based on the powertrain properties and configuration. The battery properties include the selected cell's voltage, resistance, and capacity. Studying these properties can help assess the feasibility of a given powertrain because the results show whether the powertrain can achieve a particular power output and range. Finally, analyzing the DOH's effect highlights the tradeoff between a higher electric energy ratio and range.

The power requirements of the aircraft model depend on the properties of the reference aircraft geometry, mission input, and the powertrain's energy source(s). In the fully electric configuration, the battery is the sole power provider. The range capability thus depends on the energy density, capacity, and internal resistance of the battery pack. In the hybrid electric configuration, the aircraft's power requirements are fulfilled by two sources, the battery pack and the ICE. Two methods of hybrid energy distribution are studied: ICE-only cruise and total mission hybridization. The energy distribution is varied to quantify its effects on the range while all other variables are constant.

As reviewed, some researchers use numerical modelling techniques intended for UAV technology. Other researchers have completed modelling and simulation for aircraft, but few have studied LSAs. Since LSAs are one of the aircraft types in which electrification

technologies will first be widely implemented [15], more extensive research must be conducted in the field. The models presented in this study present further novelty as the powertrain constraints are the same as a physical electrified ground-based propulsion test stand at the National Research Council of Canada. Using the test stand's constraints will allow for comparing theoretical and experimental results. However, the components in the powertrain models were assumed to have perfect efficiency in gathering the initial simulation results in this paper, thus causing discrepancies between the simulation and test stand results.

The chosen objectives to be analyzed using the various powertrain models will be instrumental in the future of sustainable aviation because the potential of electrified aircraft will be thoroughly analyzed. Although the scope of this study only pertains to LSAs, the results can be helpful for larger aircraft and future battery capabilities.

## 2. Methods

After considering the modelling techniques used in the literature, the modelling approach used in this study is selected to be based on those reported in [12,25]. These studies modelled the propulsion systems and controlled the parameters using a MAT-LAB/Simulink environment. They also used *X-Plane* to provide the necessary inputs. This study solely uses the MATLAB/Simulink environment and obtains inputs from a reference aircraft body. A thorough explanation of the methods used in this study is detailed in this section.

### 2.1. Reference Aircraft

The aircraft on which the model's geometry and weight distribution is based is a Pipistrel Virus 912. This reference aircraft was selected for its size and power output typical of LSAs. It has an MTOM of 600 kg, a true cruise airspeed of 246 km/h, a range of 1650 km, and is powered with a Rotax 912 ICE [26].

The known geometry of this aircraft is used to calculate drag and power requirements at different altitudes and over a range of flight speeds. These curves, as seen in Figures 2 and 3, are used in the flight-performance model. The drag data are calculated using a drag build-up method [27] and are fed as reference data for the model. The power data are calculated by multiplying drag by velocity and determining if the model calculations are correct. Other parameters such as MTOM, empty weight, and fuel volume are used as inputs to the model.

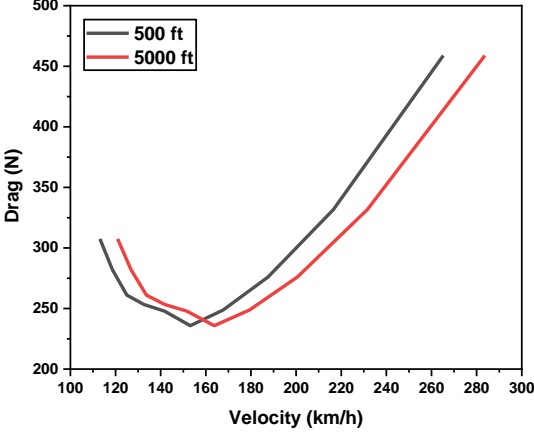

**Figure 2.** Drag vs. velocity for 500 and 5000 ft altitudes.

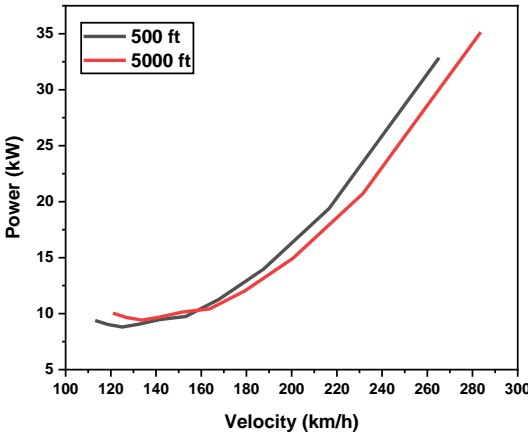

**Figure 3.** Power vs. velocity for 500 and 5000 ft altitudes.

### 2.2. Aircraft Modelling

The aircraft model's main task is to continually analyze the motor's power draw. This analysis is performed by updating the aerodynamic and powertrain parameters throughout each segment of the selected mission. In the fully electric configuration, the motor's power draw assists in determining the remaining SOC at a given time. In the hybrid powertrain configuration, the magnitude of the motor's power draw determines how it will be divided between the two sources to complete the mission effectively.

The aircraft model uses five controllers modelled after a basic PID control structure, as seen in Figure 4. Each controller has a plant block with the necessary equation of motion. An overview of the aircraft model control structure is depicted in Figure 5. A PID controller determines "error" from known reference data and the "actual" data, which are continuously fed back to calculate the error. PID control loops may also be used as embedded loops, meaning that the outer loop's plant output is used as reference data to the inner loop. The first three embedded control loops, controllers 1–3, are responsible for controlling motion in the body reference frame and represent control of motor torque, propeller angular speed, and motor thrust, respectively. The last two embedded control loops, controllers 4 and 5, are responsible for controlling motion in the ground reference frame and represent control of aircraft climb/descent angle and vertical velocity, respectively. Each plant block solves an equation of motion related to one of the five control variables. These equations of motion and their required variables/reference data are detailed in the subsequent sections. The energy distribution and variable cruise algorithms provide inputs to the five controllers.

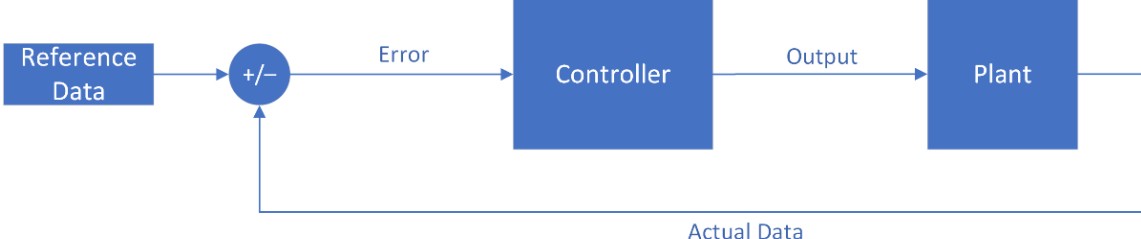

**Figure 4.** Basic PID control structure.

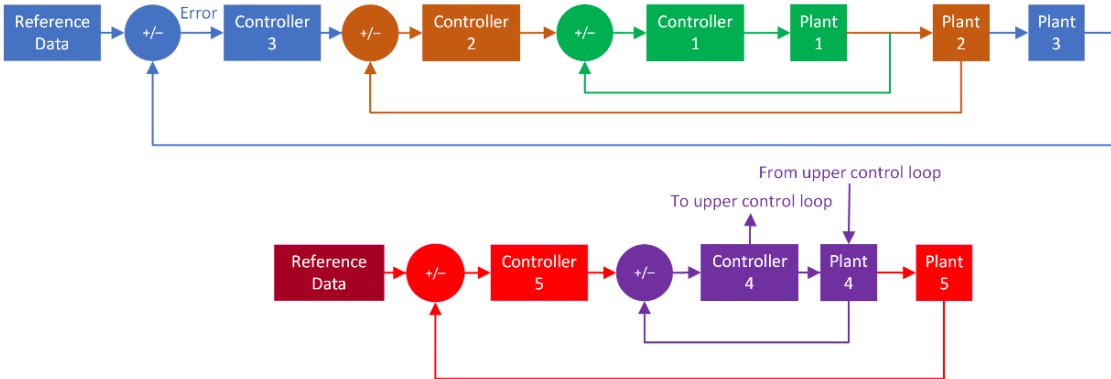

**Figure 5.** Aircraft model control structure overview.

2.2.1. Control Loop 1: Motor Torque

Controller 1 controls the electric motor's torque by inputting a reference and actual value of motor shaft rotational speed into the PID controller 1. The motor's torque-speed limits are considered by analyzing whether the demand is higher than the limit. If the motor cannot meet the demand, it outputs its maximum torque value. Finally, Plant 1 determines the motor shaft rotational speed using an equation to obtain the angular acceleration of the propeller. The angular acceleration is then integrated to obtain the desired value of rotational speed. The input of net torque is required to obtain the angular acceleration and can be determined using the following equation:

$$Q_{\text{net}} = \eta_{\text{propeller}}[(Q_{\text{motor}} \times GB) - Q_{\text{res}}] \tag{1}$$

where $Q_{\text{motor}}$ is motor torque, $GB$ the gearbox ratio, $Q_{\text{res}}$ resistive torque on the propeller, and $\eta_{\text{propeller}}$ the propeller efficiency. The experimentally determined gearbox ratio of 1.5 was chosen because the maximum motor output torque can be achieved without surpassing the propeller speed limitations. Propeller efficiency and resistive torque are determined in control loop 2 and are inputs to Plant 1.

Newton's Second Law of Torques can now be applied to calculate the propeller's angular acceleration, integrated to obtain the desired rotational speed.

$$\alpha_{\text{propeller}} = \frac{Q_{\text{net}}}{i_{\text{motor}} + i_{\text{propeller}}} \tag{2}$$

where $\alpha_{\text{propeller}}$ is the angular acceleration of the propeller, $i_{\text{motor}}$ is the electric motor inertia, and $i_{\text{propeller}}$ is the propeller inertia. $i_{\text{propeller}}$ is 0.42 kg m$^2$, the inertia value of the Virus 912's propeller [28], and $i_{\text{motor}}$ is also taken to be 0.42 kg m$^2$ because it is unknown.

2.2.2. Control Loop 2: Propeller Angular Speed

Controller 2 controls the propeller angular speed using reference inputs and actual aircraft thrust. It is tuned with the appropriate gains to achieve a rapid and precise response. Plant 2 determines three values at once: aircraft thrust, aircraft torque, and propeller efficiency. The required values are found through relationships relating to other previously known properties such as propeller speed and geometry. Propeller thrust is determined through the following equation:

$$T = k_{\text{T}} \rho \omega^2 d^4 \tag{3}$$

where $T$ is propeller thrust, $k_{\text{T}}$ is the thrust coefficient, $\rho$ is the air density, $\omega$ is the angular speed of the propeller, and $D$ is the propeller diameter. The thrust will be used as feedback into controller 3.

Propeller torque is calculated using the following equation:

$$Q_{propeller} = \frac{k_P \rho n^2 d^5}{2\pi} \tag{4}$$

where $Q$ is the propeller torque and $k_P$ is the power coefficient. The torque coefficient $k_Q$ is typically used in this equation but is substituted with $k_Q = k_P/2\pi$ for the purpose of the model. This torque and propeller efficiency are used to calculate the net torque in Plant 1, as previously described. The propeller efficiency, thrust coefficient, and power coefficients are functions of propeller design, rotational speed, and flight velocity. Relationships between these values and the propeller's advance ratio ($J$) were experimentally determined in [29] and are valid for all 2, 3, and 4 bladed propellers. These values are used as reference data in this control loop.

### 2.2.3. Control Loop 3: Aircraft Thrust

Controller 3 uses a reference velocity from the selected mission profile and the actual velocity output from Plant 3 to determine the error input required by the PID controller. The PID then accurately controls the thrust demand through tuning for precise tracking of the reference mission profile, despite the quick velocity transitions throughout the profile. Plant 3 calculates flight velocity by integrating the acceleration value found using fundamental laws of motion. Mass, drag, thrust, weight, and climb/descent angle are the inputs to the equation as seen below:

$$ma = T - D - W \sin \gamma \tag{5}$$

where $m$ is aircraft mass, $a$ is acceleration, $T$ is thrust, $D$ is drag, $W$ is weight, and $\gamma$ is the climb/descent angle. The total thrust and the climb/descent angle are inputs from other control loops, while the mass, weight, and drag were previously determined based on the reference aircraft parameters, which can be called upon as needed.

### 2.2.4. Control Loop 4: Climb/Descent Angle

Control loops 4 and 5 utilize the ground reference frame because the control variables are related to the aircraft's relative position to the ground. The fourth controller controls the aircraft's climb/descent angle using the aircraft's vertical velocity error and the PID. Plant 4 uses the same equation as Plant 3, but forces act only vertically. The sum of vertical forces equation is as follows, where it is desired to find vertical acceleration:

$$ma_z = T \sin \gamma + D \sin \gamma + L \cos \gamma - W \tag{6}$$

where $L$ is lift. The lift was assumed to be equal to weight for this study as the climb/descent angles are minimal during the mission.

### 2.2.5. Control Loop 5: Vertical Velocity

Controller 5 uses the error of altitude values, a reference value for a predetermined profile, and an actual value for Plant 5 to control the vertical velocity. Plant 5 simply integrates the vertical velocity from Plant 4 to obtain the vertical position (i.e., altitude).

### 2.3. Powertrain Energy Distributions

As described in the literature review section, there are several ways that current studies of hybrid systems have distributed the system's power requirements between the two hybrid energy sources. Some of these studies included the ideal operating line method [15,18], the constant DOH method [11], and the rule-based controller method [12]. The selected method is similar to the rule-based controller in [12], but it is based on the maximum continuous power output of the ICE, as will be described below.

The first energy distribution module tested simulates the fully electric powertrain. Since the fully electric aircraft only has one energy source, its battery pack can only dis-

charge, as no energy recovery was considered for this study. Thus, the only calculation required is the SOC of the battery at a given time. First, the electric motor power draw is easily calculated by multiplying two of the main aircraft control structure outputs, motor torque and motor shaft angular speed. Next, the power draw is divided by measured voltage to obtain the current. Finally, the current is used to calculate SOC using the following equation:

$$SOC = \int_0^t \frac{I_{\text{pack}}}{C_{\text{pack}}} \, \mathrm{d}t \tag{7}$$

where $I_{\text{pack}}$ is the battery pack's current and $C_{\text{pack}}$ is the battery pack's capacity. The SOC is then used to calculate the open-circuit voltage and resistance via reference data of battery properties.

The hybrid electric module calculates the SOC in the same way as the fully electric, but the motor's power draw must first be distributed between the two sources: the ICE and the battery pack. There are two methods of distribution explored in this paper: ICE-only cruise method and the total mission hybridization method. Both methods involve an energy distribution algorithm, but the conditions are slightly different.

The ICE-only cruise algorithm has four main conditions that it obeys. The first condition is applied when the motor's power draw is larger than the ICE's maximum power output. Therefore, as long as the battery pack has an ample SOC, the ICE operates at its maximum power output, and the battery will produce the additional required power. The second condition pertains to when the motor's power draw is less than the maximum power of the ICE, and the battery has a sufficient SOC. In this case, the selected DOH will be utilized to calculate the desired power distribution. The third condition states that when the aircraft is flying during its cruising phase, the ICE will generate all power required. Lastly, the fourth condition, a safety condition, checks that SOC does not fall below 20%. Discharging below this level can have lasting effects on battery health [30]. Thus, the safety condition is critical to preserve the lifetime of the powertrain and reduce the rate of degradation [31]. Therefore, when the SOC reaches 20%, the powertrain will be entirely powered by the ICE to ensure that the safety condition is upheld.

The total mission hybridization algorithm only has three conditions corresponding to conditions 1, 2, and 4 from the previous description. The only difference is that condition 2, involving hybridization based on DOH, will be the selected condition for most of the mission, including the extensive cruise phase. Thus, the only time the powertrain will not be operating in a variation of a hybrid mode is in the emergency case where the SOC of the battery becomes less than 20%.

Once the power output value of each energy source is determined at a given time, the SOC can be determined using Equation (7), and the fuel consumption can be determined using reference data sets. These data sets refer to the fuel consumption of the ICE, the Rotax 912, at a particular power output and rotational speed.

### 2.4. Variable Cruise Algorithm

The range is maximized through a second algorithm that ensures that the mission's climb and descent velocities, altitudes, and times remain constant. However, the cruise time is variable and dependent on the remaining energy of the powertrain. The cruise length is dependent on a trigger point based on the remaining energy of the source or sources. Since there is only one energy source in the fully electric powertrain, the battery pack, the trigger point is dependent on the SOC only. However, there are two energy sources in a hybrid configuration, meaning that the trigger point depends on the SOC and the remaining fuel volume. The algorithm does not leave any fuel reserves since the maximum range is desired, and only leaves the previously mentioned 20% SOC for safety purposes. After the end of the cruise segment, the descent and landing segments will begin. The trigger point can be updated, so a safe landing is always achieved based on the mission and the initial energy of the powertrain.

## 2.5. Aircraft Mass Analysis

A mass analysis gives the maximum mass of components unique to electrified powertrains, such as batteries. Most components found on the traditionally powered aircraft are still required in the electrified systems, such as its structure, instrumentation, crew, and engine. The hybrid configuration's selected ICE and fuel volume determine the allowable mass of batteries and associated components used in the hybrid powertrain. This difference is because fuel volume and ICE selection are the only varying masses between fully electric and hybrid electric configurations. A significant distinction between a hybrid electric and a traditional configuration is the addition of electrical components such as the battery pack, electric motor, inverter, and cables. Additionally, there is a decrease in fuel volume in the hybridized system since the battery will produce a fraction of the required power. Independently of the propulsion system, the MTOM of the aircraft was set to 600 kg for the herein discussed study.

The traditional ICE and structures are entirely removed and replaced by batteries, the electric motor, motor controller, and miscellaneous components such as wires and battery casing for a fully electric configuration. In electric vehicles, the mass of these miscellaneous components is approximately 25% of the total battery mass [32]. Therefore, this value was assumed for the fully electric configuration and was increased to 35% for the hybrid electric configuration to account for the additional hybrid components.

A mass breakdown for each powertrain configuration is presented in Table 1. The empty mass, in this case, is the empty mass of the Virus 912 less the mass of its power plant, as it will not be used in the fully electric configuration. However, this empty mass still includes the onboard equipment typically included in the empty mass. The other powertrain components are off-the-shelf, meaning their masses are known [26,33–36].

**Table 1.** Traditional, fully electric, and hybrid electric powertrain aircraft mass breakdowns.

| Component | Traditional Masses (kg) | Fully Electric Masses (kg) | Hybrid Electric Masses (kg): 20 kg Fuel | Hybrid Electric Masses (kg): 10 kg Fuel |
|---|---|---|---|---|
| Empty Mass | 211.6 | 211.6 | 211.6 | 211.6 |
| Engine/Generator | 75.4 | | 75.4 | 75.4 |
| Fuel | 48.5 | | 20 | 10 |
| Maximum Battery Mass | | 150.7 | 84 | 91.4 |
| Wiring | | 37.68 | 29.4 | 32 |
| Electric Motor | | 42.5 | 42.5 | 42.5 |
| Motor Controller | | 7.5 | 7.5 | 7.5 |
| Crew | 150 | 150 | 150 | 150 |
| Payload | 114.5 | | | |
| **Total** | | **600** | | |

The hybrid electric configuration will utilize an ICE/generator of the same mass and power capability as the Rotax engine used in the traditional configuration. A constant fuel mass of either 10 kg or 20 kg was selected for the simulations. Knowing the fuel mass assumptions, one can determine the maximum allowable battery masses for each fuel mass case.

The maximum allowable battery mass was found to be 150.72 kg for fully electric, 84.0 kg for hybrid electric (20 kg fuel), and 91.4 kg for hybrid electric (10 kg fuel), respectively. For the sake of the mass analysis, no payload was budgeted for either electrified powertrain to ensure that the maximum allowable battery mass is known when determining the pack configurations. Using the maximum allowable battery mass is necessary because with an already limited mass budget, to analyze aviation electrification feasibility thoroughly with a typical lithium-ion battery's energy density, every gram counts. Once the pack configurations are determined, any additional mass reserved for batteries can be used for payload at that time or can be taken as an MTOM decrease.

*2.6. Simulation Parameters*

As previously described, feasibility was studied by modifying three powertrain parameters (cruising altitude, battery properties, and DOH) in various combinations. A variable mission profile composed of five segments was tested. Only the cruise segment is variable, as previously mentioned. The segments were as follows:

1. Takeoff and initial climb to 152.4 m (500 ft) at 100% power.
2. Continued climb at the best rate of climb until the desired cruising altitude is reached.
3. Cruise at a constant true airspeed of 246 km/h.
4. Descending flight (maintaining cruise speed) at a constant descending rate.
5. Approach and landing.

The cruising altitudes tested using the models are 762 m (2500 ft), 1524 m (5000 ft), and "maximum altitude". The maximum altitude depends on the voltage of the given battery pack because the electric motor's torque-speed limits are dependent on the voltage. Therefore, operating a given powertrain model using a battery pack with a lower nominal voltage has a reduced torque-speed limit and thus a reduced power output. If the excess power required to climb to a given altitude exceeds the maximum excess power provided by the electric motor, the powertrain will be unable to complete this climb. The maximum altitudes for the various battery packs used in this study were found to be between 1524 m (5000 ft) and 2591 m (8500 ft), which are typical cruising altitudes for LSAs.

As calculated for the fully electric and hybrid electric configurations in the previous section, the available mass for batteries was used to determine the chemistry and number of cells used in the simulations. Since it is desired to study the feasibility of implementing hybrid or fully electric propulsion systems in aircraft in the next five years, the selected battery will likely use lithium-ion. Lithium-ion is one of the most common and most studied battery chemistries. Moreover, it has been the one to see the most success in similar electrification applications such as electric vehicles.

A commercially available lithium-ion battery was chosen as the first battery tested in the aircraft model. The LG Chem is a pouch-type cell used in electric vehicles such as the Ford Focus Electric 2017 [35]. Each cell has a nominal voltage of 3.7 V, an energy density of 111 Wh/kg, a mass of 703 g, and a nominal capacity of 21 Ah [35]. The current state-of-the-art energy densities available for electrified aircraft are often higher than the selected cell for this study, which was chosen due to the availability of the battery specifications. Although using state-of-the-art energy density values would provide slightly longer range results, the trends of the overall results remain the same.

The cell array configuration was modified to suit the needs of the aircraft model's mass and voltage requirements. The nominal pack voltage must be between 400–800 $V_{DC}$, which is the voltage range of the electric motor of the ground-based test stand [36]. Voltage can be increased by adding more cells in series in the pack design. The battery pack configurations used in the fully electric and hybrid electric simulations are listed in Table 2 below, all of which only use one module in parallel due to mass restrictions. Adding battery modules in parallel increases the pack capacity, but the maximum battery mass determined in the mass analysis can only accommodate one module in parallel. Since each configuration only uses one module in parallel, the capacity of each will be the same, meaning that voltage/number of cells is the primary differentiating property that will affect range.

**Table 2.** Selected battery pack configurations.

| Battery Pack | Type of Powertrain | Cell Configuration | Nominal Voltage |
|:---:|:---:|:---:|:---:|
| 1 | Fully Electric | 214 S, 1 P | 800 V |
| 2 | Fully Electric | 190 S, 1 P | 700 V |
| 3 | Fully Electric | 163 S, 1 P | 600 V |
| 4 | Hybrid Electric | 130 S, 1 P | 480 V |
| 5 | Hybrid Electric | 119 S, 1 P | 440 V |

The DOHs selected for analysis were 0.5 and 0.3, where 0.5 is an even distribution of the power demand between the energy sources, and 0.3 is a case using less battery power and more ICE power. Both will provide insight into the advantages of hybrid electric propulsion, such as fuel burn reduction. Due to the relatively low capacity of the selected batteries, analyzing a higher DOH would not be beneficial as the SOC would diminish very quickly.

## 3. Results and Discussion

The simulation models are run using various combinations of the simulation parameters to study their influence on the range and overall feasibility. The fully electric model and the hybrid electric models with the two different energy distribution strategies are analyzed separately. It is found that the comprehensive result themes are comparable for all models.

### 3.1. Fully Electric Simulation Results

A maximum and minimum range of 92.2 km and 64.8 km, respectively, was calculated from the fully electric model. Compared to the reference aircraft powertrain using a traditional ICE, these range values are immensely low. Even the maximum fully electric range is only 5.6% of the traditionally powered reference aircraft range of 1650 km. The low capacity and high mass qualities of the cells used are the leading cause of the low range. Additionally, the 600 kg mass limitation of the LSA reference body hinders the range further as the number of cells used is already at its maximum. Despite the low range, there are trends in battery nominal voltage, cruising altitude, and aircraft mass to be studied.

Figure 6 shows the effect of cruise altitude and battery pack voltage on the range. Unsurprisingly, increasing the nominal voltage (overall energy storage) affects the range. The range increases with increasing voltage for a constant altitude due to the direct relationship between power and voltage. Thus, increasing battery nominal voltage increases power output, leading to an overall increase in the range since power demands will remain constant for a particular mission. Using multiple 800 V modules in parallel could also increase range, but unfortunately, it is not possible due to weight restrictions. The range is also improved with increased cruise altitude because of the reduced drag and power draw when flying at the same true airspeed of 246 km/h at greater altitudes, as is shown in Figures 2 and 3. Unfortunately, the maximum altitude for a given battery pack due to the previously mentioned torque-speed limitations prevents cruising any higher to improve range further. Thus, there are no data in the higher altitude–lower voltage region of Figure 6. Figure 6's trendlines highlight that both altitude and voltage impact the overall range through a direct relationship. Thus, a low altitude–high voltage combination and a high altitude–low voltage combination will result in the same range. As expected, the maximum range is at the maximum altitude and maximum voltage point.

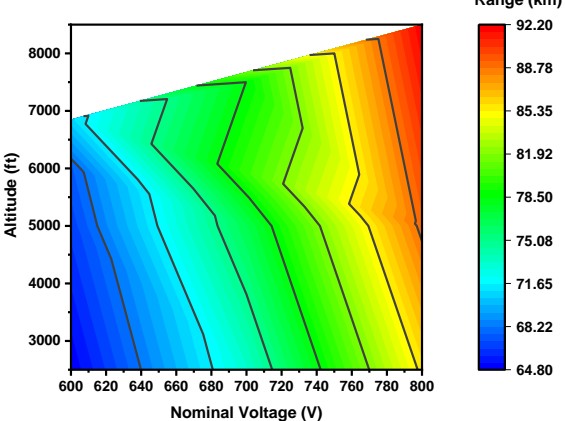

**Figure 6.** Fully electric altitude vs. voltage range comparison.

Figure 7 shows the effect that takeoff mass has on the overall range. Mass only has a minor, though inversely proportional, influence on the range. An increased mass requires additional power during the climb, thus requiring additional battery power to achieve it. Figure 7's trendlines show that achieving the same range using a high mass–high voltage or a low mass–low voltage combination is possible because of the inversely proportional relationship of aircraft mass and range. However, mass's effect on the range only begins to be seen at a mass of 585 kg. This trend could be due to the relationship between aircraft mass and nominal voltage because of the mass limit of the standard LSA. The 800 V battery represents the maximum battery mass accounted for in the mass analysis, meaning that these cases are at the MTOM. Using a 600 V or 700 V battery, there are two options: take no payload and decrease the MTOM or take a payload corresponding to an MTOM of 600 kg. This option is why Figure 7 only contains data in the top half of the plot.

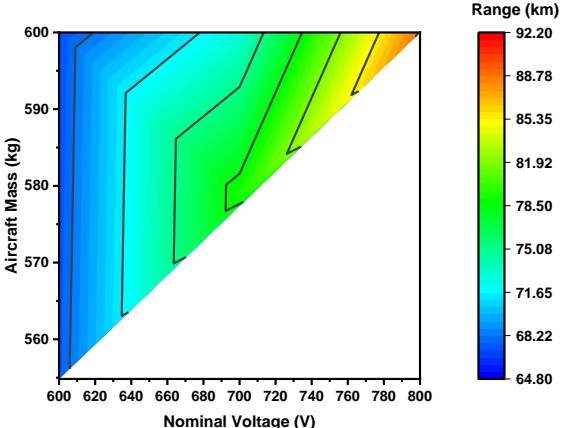

**Figure 7.** Fully electric mass vs. voltage range comparison.

### 3.2. Hybrid Electric Results (ICE-Only Cruise)

The ICE-only cruise distribution method in the hybrid electric model has a maximum range of 728 km and 376.9 km using 20 kg and 10 kg of fuel, respectively. These values are 44.1% and 22.8% of the traditionally powered reference aircraft range and are achieved using 41.2% and 20.6% of the reference aircraft's fuel mass of 48.5 kg. Although the range is substantially reduced compared to the reference aircraft, fuel usage has experienced a slightly more significant decrease. The benefit of hybrid propulsion is seen in analyzing the range to fuel mass ratios. The results show that hybrid powertrains use a reduced fuel fraction overall. However, a significant difference between the traditional and hybrid powertrain's maximum ranges is seen. The hybrid system cannot complete the mission if a required flight exceeds 728 km. Thus, the hybrid range will not increase until the MTOM and/or battery energy density increases over time.

Figures 8 and 9 depict that the total fuel mass significantly affects the total range. This relationship was expected because of the low energy density of current battery technology compared to fuel mass, as previously discussed. Additionally, due to the mass restrictions, it was difficult to make up for this low energy density in the pack design, meaning that the pack has limited capability to provide power to the system. However, apart from the fuel mass, the plots of the hybrid model simulation results can also show valuable trends elsewhere. Although these other parameters have a nearly insignificant range influence compared to fuel mass, they will become more prominent with technological advancement. Therefore, it is essential to understand their effect. For example, the trendlines of Figure 9 show that the altitude also influences range due to lower air density and power requirements, as previously described. However, altitude's influence is not quite as impactful as fuel mass is.

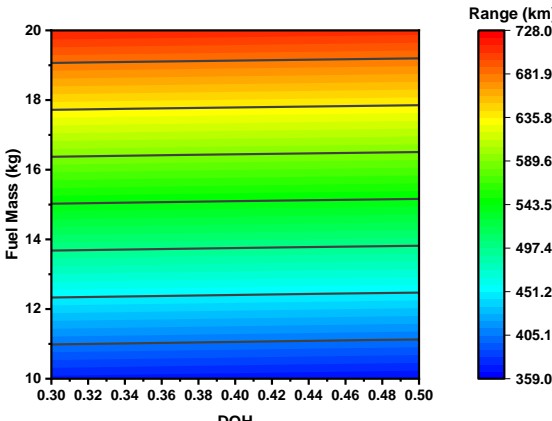

**Figure 8.** Hybrid electric fuel mass vs. DOH range comparison (ICE-only cruise).

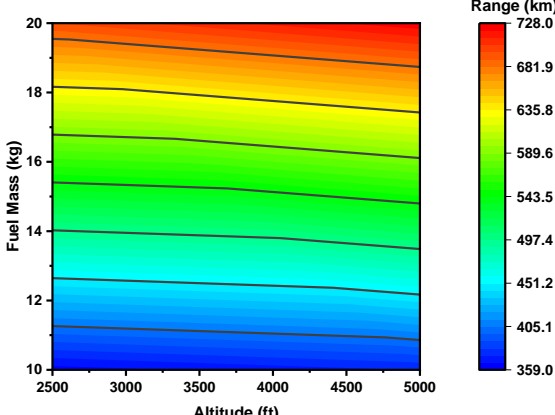

**Figure 9.** Hybrid electric fuel mass vs. altitude range comparison (ICE-only cruise).

In Figures 8 and 9, it is not easy to envision the effects of DOH and battery nominal voltage on range due to the significant influence of fuel mass. Therefore, plots showcasing only the 10 kg fuel hybrid simulation results are seen in Figures 10 and 11. These segregated result plots allow for the effects of DOH and nominal battery voltage to be visualized accurately by removing the overwhelming fuel mass impact.

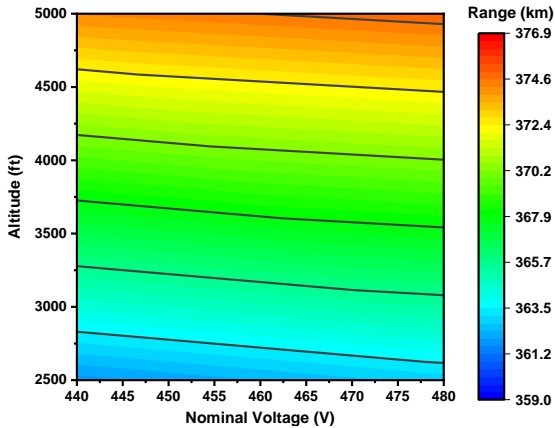

**Figure 10.** Hybrid electric altitude vs. voltage range comparison: 10 kg cases only (ICE-only cruise).

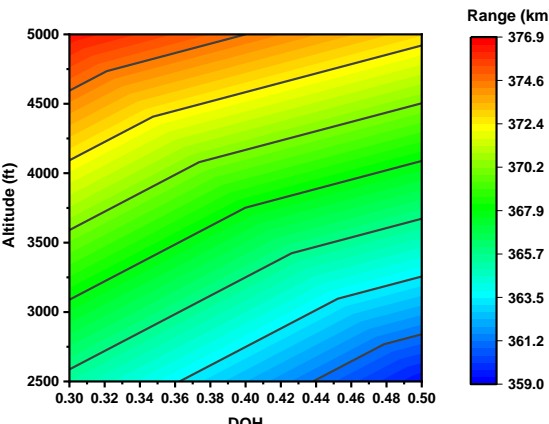

**Figure 11.** Hybrid electric altitude vs. DOH range comparison: 10 kg cases only ((ICE-only cruise).

Figure 10 shows that an increase in nominal voltage only slightly affects the range. These results differ from the fully electric results, where it was found that nominal voltage is one of the leading parameters affecting the range. Reducing fuel mass to increase the number of cells (therefore increasing nominal voltage) makes a minimal difference in the range since the power produced by the ICE, even with less fuel, is much greater than the power provided by the battery pack. From the point of view solely focused on increasing range, reducing fuel mass to increase the number of cells is undesirable because the fuel mass has a much more significant effect on the overall range.

Figure 11 shows that reducing DOH provides a more extended range. Since the battery power has little influence compared to the ICE power, distributing the battery energy over a more significant percentage of the mission is more beneficial. Trendlines highlight that high altitude–high DOH or low altitude–low DOH combinations produce the same total range due to the inversely proportional effect of the DOH. Therefore, the maximum range is achieved at the maximum altitude and minimum DOH.

*3.3. Hybrid Electric Results (Total Mission Hybridization)*

The aim of the simulations using the total mission hybridization energy distribution method is slightly different from that of the two previous results sections. The range capabilities of the hybrid powertrain were seen in Section 3.2, as were the range comparisons to a traditionally powered aircraft of the same MTOM. In this section, a similar comparison will be made, but it will be between the two energy distribution methods rather than the traditionally powered aircraft. This comparison will show whether the energy distribution method impacts the range when the fuel mass, battery configuration, and mission remain constant.

As previously mentioned, the model's variable cruise algorithm ensures that range is maximized by leaving the cruise time variable and is dependent on the remaining energy of the powertrain. The ICE-only cruise method ensures that both energy sources are depleted to their safe levels upon reaching the trigger point, thus maximizing range. This method uses higher DOH values (30% and 50%), which is made possible since the battery is only being used during climb and descent. Both sources are continually depleting in the total mission hybridization method, meaning that whichever one reaches its trigger point first will initiate the landing sequence. If the selected constant DOH is too high, the battery will deplete too quickly, and there will be excess fuel upon landing. If it is too low, the fuel reserve will deplete before the battery, thus wasting potential range capability. This study also shows the optimal constant DOH for a given altitude/battery combination.

The range results of 24 simulations with varying DOH and altitude/battery combinations are shown in Tables 3 and 4. Altitude 1 is 5000 ft and altitude 2 is 2500 ft, while the battery data were previously listed in Table 2. The fuel mass is kept constant at 10 kg for all simulations. Table 3's change in range value is compared to the same simulation run

for the ICE-only cruise case with a 50% DOH, while Table 4's is compared to the ICE-only cruise case with a 30% DOH. The yellow highlighted cells are the most significant change in range cases.

**Table 3.** Total mission hybridization simulation results compared to 50% DOH ICE-only cruise simulations.

| Battery 4, Altitude 1 | | | Battery 5, Altitude 1 | | | Battery 4, Altitude 2 | | | Battery 5, Altitude 2 | | |
|---|---|---|---|---|---|---|---|---|---|---|---|
| DOH (%) | Range (km) | ΔRange (km) | DOH (%) | Range (km) | ΔRange (km) | DOH (%) | Range (km) | ΔRange (km) | DOH (%) | Range (km) | ΔRange (km) |
| 10 | 378.08 | 5.00 | 10 | 378.07 | 5.52 | 10 | 369.69 | 9.88 | 10 | 369.68 | 10.68 |
| 11 | 382.24 | 9.16 | 11 | 382.23 | 9.68 | 11 | 369.73 | 9.91 | 11 | 369.72 | 10.72 |
| 12 | 385.05 | 11.98 | 12 | 385.05 | 12.49 | 12 | 369.76 | 9.95 | 12 | 369.76 | 10.75 |
| 13 | 385.53 | 12.46 | 13 | 378.53 | 5.97 | 13 | 369.80 | 9.99 | 13 | 354.14 | −4.87 |
| 14 | 383.67 | 10.60 | 14 | 353.59 | −18.97 | 14 | 358.98 | −0.83 | 14 | 330.88 | −28.12 |
| 15 | 360.05 | −13.02 | 15 | 331.97 | −40.58 | 15 | 336.95 | −22.86 | 15 | 310.73 | −48.27 |

**Table 4.** Total mission hybridization simulation results compared to 30% DOH ICE-only cruise simulations.

| Battery 1, Altitude 1 | | | Battery 2, Altitude 1 | | | Battery 1, Altitude 2 | | | Battery 2, Altitude 2 | | |
|---|---|---|---|---|---|---|---|---|---|---|---|
| DOH (%) | Range (km) | ΔRange (km) | DOH (%) | Range (km) | ΔRange (km) | DOH (%) | Range (km) | ΔRange (km) | DOH (%) | Range (km) | ΔRange (km) |
| 10 | 378.08 | 1.23 | 10 | 378.07 | 2.08 | 10 | 369.69 | 3.70 | 10 | 369.68 | 5.05 |
| 11 | 382.24 | 5.39 | 11 | 382.23 | 6.24 | 11 | 369.73 | 3.74 | 11 | 369.72 | 5.09 |
| 12 | 385.05 | 8.20 | 12 | 385.05 | 9.05 | 12 | 369.76 | 3.78 | 12 | 369.76 | 5.12 |
| 13 | 385.53 | 8.68 | 13 | 378.53 | 2.54 | 13 | 369.80 | 3.82 | 13 | 354.14 | −10.50 |
| 14 | 383.67 | 6.82 | 14 | 353.59 | −22.41 | 14 | 358.98 | −7.00 | 14 | 330.88 | −33.75 |
| 15 | 360.05 | −16.80 | 15 | 331.97 | −44.02 | 15 | 336.95 | −29.03 | 15 | 310.73 | −53.90 |

It was found that the DOH to achieve optimal range is 13% using battery 1 and 12% using battery 2. Values of DOH less than the optimal still show an increased range compared to the ICE-only cruise missions (positive change in range). The fuel reserve has been depleted upon landing with a less than optimal DOH, but some battery energy remains. The range rapidly decays with decreasing DOH for the altitude 1 cases, while the altitude 2 cases show a more gradual decay. This trend is likely because altitude 2 requires less energy to land once the variable cruise algorithm's trigger point has been reached since it is representative of the lower altitude.

Values of DOH greater than the optimal values cause a sharp decrease in range for all combinations of battery and altitude. In these cases, the battery has depleted upon landing, but there is still excess fuel, diminishing the range prospect. Since fuel impacts the range more significantly than the battery, excess fuel will be more detrimental to the range than excess battery SOC.

It should also be noted that the change in range values in Table 3 seem to be significantly larger than those of Table 4. As described in the previous section, an ICE-only cruise mission had a slightly longer range using 30% DOH compared to 50% DOH. Therefore, of the three hybrid powertrain energy distributions tested, the best range to worst range is as follows: total mission hybridization (using optimal DOH value), ICE-only cruise (using 30% DOH), ICE-only cruise (using 50% DOH), where the total mission hybridization energy distribution method can provide ranges up to 3% longer than the other methods. Although this is not a vast improvement, it shows some influence of the energy distribution method on the range.

## 4. Conclusions

Fully electric and hybrid electric simulation results have highlighted the pros and cons of electrified propulsion in the present-day aviation industry. Both configurations achieved the mission profile velocities and altitudes required with a prompt control response. Although the fully electric range was only about 5% of the range of a traditionally powered reference aircraft, the results nevertheless emphasized trends within other simulation parameters. The hybrid powertrain ranges are more comparable to the reference aircraft range. However, the majority of this range capability results from the superior energy density of fuel mass, not the contribution of the battery pack to the system. It is anticipated that fuel mass will continue to have a heightened effect on range, even as battery energy density is increased over time. However, the fraction of range achieved through battery power will have progressed slightly.

Results also showed that the energy distribution method impacts the total range when the fuel mass and battery configuration remain constant. Using the total mission hybridization method, a range of up to 3% longer than the same mission is possible. It was found that the optimal DOH for this type of energy distribution is 13% for flights at 5000 ft and 12% for flights at 2500 ft.

To conclude, electrified propulsion systems of fully electric and hybrid electric configurations are not yet feasible to be widely implemented in LSAs. The current battery technology limits the systems' range capabilities, which are not up to the conventional standard of traditionally powered LSAs. However, when utilizing more batteries and exiting the LSA regime, it is possible to achieve similar ranges of traditionally powered LSAs, but it comes at a financial and MTOM cost. The challenge in this aircraft segment is that the MTOM limit of LSA significantly restricts the size of the battery pack. As a result, most benefits of electrification are not realized. However, there are future battery technologies in development that are likely to increase energy densities in the lithium-ion category and even more significantly using other battery chemistries. Implementing these future technologies will increase electrification feasibility and major pollutant reduction in the aviation industry.

**Author Contributions:** Conceptualization, A.E.K. and O.A.; methodology, M.M., A.E.K., G.B. and E.D.; software, M.M. and G.B.; validation, M.M. and G.B.; formal analysis, M.M.; investigation, M.M., resources, O.A.; data curation, M.M.; writing—original draft preparation, M.M.; writing—review and editing, A.E.K., G.B. and O.A.; visualization, M.M.; supervision, A.E.K. and G.B.; project administration, A.E.K.; funding acquisition, A.E.K. All authors have read and agreed to the published version of the manuscript.

**Funding:** The authors thank the National Research Council of Canada for funding under contract No. 954291 awarded to Ahmet E. Karataş for the support of this research work.

**Institutional Review Board Statement:** Not applicable.

**Informed Consent Statement:** Not applicable.

**Data Availability Statement:** The data that support the findings of this study are available from the corresponding author, A.E.K., upon reasonable request.

**Conflicts of Interest:** The authors declare that they have no known competing financial interests or personal relationships that could have appeared to influence the work reported in this paper.

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
