# Peer review of "Feasibility Study of Electrified Light-Sport Aircraft Powertrains†"

_aerospace, doi:10.3390/aerospace9040224_

Round 1

Reviewer 1 Report

The paper describes the simulation of hybrid-electric powertrains for an LSA type application. The simulation loops in a MATLAB/Simulink environment are well described and realistic conclusions are drawn from the results. Here a small number of comments to the work:

line 18 (minor): reference is made in the abstract to a physical test stand, but its connection to the simulation models does not become clear in the text.

lines 46ff (minor):
(a) the difference between "air passenger traffic" and "passenger kilometers" does not become clear - is "traffic" used as a synonym for number of flights? Please clarify.
(b) is air cargo traffic really "additionally"? This would mean that you look only at the number of pure cargo flights in this sentence, which looks dubitable, as a good share of freight is carreid in the belly of passenger flights.

Figure 5 (minor): low quality, white text on pink background barely legible.

line 261 (major): unit for inertia is kg x m2, not kg/m2.

equation (6) (major): the vertical lift component is L x cos(gamma), not L x sin(gamma)! Please check your algorithm accordingly.

section 2.4 (minor): it should be stated explicitely, that the algorithm does not foresee any energy reserves when calculating range.

lines 426ff (minor): the selected automotive cell is a very conservative approach, as for example the Pipistrel alpha electro has an energy density of 155 Wh/kg on systems level. Li ion cells today are around 180-250 Wh/kg, leading to an energy density on systems level of 150-220 Wh/kg in flying applications. This somewhat distorts the results on range that can be achieved. The selection has been done, but this paragraph could merit some words on State of Art in aviation to put the calculation results in relation.

Author Response

General Comment: The paper describes the simulation of hybrid-electric powertrains for an LSA type application. The simulation loops in a MATLAB/Simulink environment are well described and realistic conclusions are drawn from the results.

Response: We would like to thank the reviewer for their detailed comments and insight.

line 18 (minor): reference is made in the abstract to a physical test stand, but its connection to the simulation models does not become clear in the text.

Response: This line has been modified for clarity.

Lines 46ff (minor):
(a) the difference between "air passenger traffic" and "passenger kilometers" does not become clear - is "traffic" used as a synonym for number of flights? Please clarify.

Response: Definitions have been added for clarification.

(b) is air cargo traffic really "additionally"? This would mean that you look only at the number of pure cargo flights in this sentence, which looks dubitable, as a good share of freight is carreid in the belly of passenger flights.

Response: This was supposed to represent all cargo, not just pure cargo flights. It has been updated for clarification in the manuscript.

Figure 5 (minor): low quality, white text on pink background barely legible.

Response: We have increased the resolution and changed the pink to red for better legibility.

line 261 (major): unit for inertia is kg x m2, not kg/m2.

Response: This has been corrected.

equation (6) (major): the vertical lift component is L x cos(gamma), not L x sin(gamma)! Please check your algorithm accordingly.

Response: We apologize for the typographical error; this has been corrected. This was only an error in the manuscript, not in the algorithm.

section 2.4 (minor): it should be stated explicitely, that the algorithm does not foresee any energy reserves when calculating range.

Response: A sentence has been added to address this.

lines 426ff (minor): the selected automotive cell is a very conservative approach, as for example the Pipistrel alpha electro has an energy density of 155 Wh/kg on systems level. Li ion cells today are around 180-250 Wh/kg, leading to an energy density on systems level of 150-220 Wh/kg in flying applications. This somewhat distorts the results on range that can be achieved. The selection has been done, but this paragraph could merit some words on State of Art in aviation to put the calculation results in relation.

Response: Thank you for your suggestion regarding the selected cell. The main reason for the selection was due to the availability of the battery specifications, as more detailed technical data is often confidential. However, a sentence has been added to address the current state of the art for readers’ knowledge.

Reviewer 2 Report

The manuscript is well written with clear descriptions of the methodology and the results. 

Author Response

General Comment: The manuscript is well written with clear descriptions of the methodology and the results. 

Response: We would like to thank the reviewer for their positive comment regarding the manuscript.

Reviewer 3 Report

In this work, the authors used theory-based model for determining the performance of a light sport aircraft in order to study the electrification and hybridization of the aircraft. The model is well defined and the results shows a good perspective in hybridization of LSA using present battery technology. The manuscript is well presented and I suggest the publication of this work in Aerospace journal. However, there are some issues and minor corrections that is better to be clarified in this manuscript.

  • The structure of control system, as presented in the manuscript, is not properly depicted by Fig. 5.
  • The authors mention the implementation of experimental data of the propeller (line278). Is there the experimental data for actual Virus 912's propeller is used or there have been an approximation using similar propeller? If the approximation method is used, the propeller should be mention in the manuscript.
  • Is equation 4 correct? Why the power coefficient is used in torque equation?
  • Minor correction:

Line 260-261: the units of mass moment of inertia are incorrect.

Table1: row 2 column 4: ‘kg’ should be removed.

Line 538: word ‘section B’ is not clear.

Author Response

General Comment: In this work, the authors used theory-based model for determining the performance of a light sport aircraft in order to study the electrification and hybridization of the aircraft. The model is well defined and the results shows a good perspective in hybridization of LSA using present battery technology. The manuscript is well presented and I suggest the publication of this work in Aerospace journal. However, there are some issues and minor corrections that is better to be clarified in this manuscript.

Response: We would like to thank the reviewer for their comments on improving the paper and for their suggestion that our manuscript be published.

The structure of control system, as presented in the manuscript, is not properly depicted by Fig. 5.

Response: The aircraft control structure algorithm is quite complex with several feedback loops. It has been a challenge to condense the data into a single figure. We believe that Figure 5 is still the best way to provide a visual depiction. To address the Reviewer’s concerns, a more detailed description was provided in Section 2.2.

The authors mention the implementation of experimental data of the propeller (line278). Is there the experimental data for actual Virus 912's propeller is used or there have been an approximation using similar propeller? If the approximation method is used, the propeller should be mention in the manuscript.

Response: The experimental data mentioned pertains to the propeller coefficient relationships determined by B. D. Hartman in reference number 29, which are valid for all 2, 3, and 4 bladed propellers. A statement was added to the manuscript to make this clearer.

Is equation 4 correct? Why the power coefficient is used in torque equation?

Response: The torque coefficient is in the equation in the form  because of the availability of the propeller’s power coefficient curve. A sentence was added to the manuscript for clarification.

Line 260-261: the units of mass moment of inertia are incorrect.

Response: This has been corrected.

Table1: row 2 column 4: ‘kg’ should be removed.

Response: This has been corrected.

Line 538: word ‘section B’ is not clear.

Response: We apologize for this error; it has been modified to “section 3.2”.

This manuscript is a resubmission of an earlier submission. The following is a list of the peer review reports and author responses from that submission.